# In Vivo Targets of *Pasteurella Multocida* Toxin

**DOI:** 10.3390/ijms21082739

**Published:** 2020-04-15

**Authors:** Arshiya Banu, Alistair J. Lax, Agamemnon E. Grigoriadis

**Affiliations:** 1Department of Microbiology, King’s College London, Guy’s Hospital, London SE1 9RT, UK; 2Centre for Craniofacial and Regenerative Biology, King’s College London, Guy’s Hospital, London SE1 9RT, UK

**Keywords:** *Pasteurella multocida* toxin, G-proteins, proliferation, QE antibody, Ki67, pHH3, β-catenin

## Abstract

Many *Pasteurella multocida* strains are carried as commensals, while some cause disease in animals and humans. Some type D strains cause atrophic rhinitis in pigs, where the causative agent is known to be the *Pasteurella multocida* toxin (PMT). PMT activates three families of G-proteins—G_q/11_, G_12/13_, and G_i/o_—leading to cellular mitogenesis and other sequelae. The effects of PMT on whole animals in vivo have been investigated previously, but only at the level of organ-specific pathogenesis. We report here the first study to screen all the organs targeted by the toxin by using the QE antibody that recognizes only PMT-modified G-proteins. Under our experimental conditions, short-term treatment of PMT is shown to have multiple in vivo targets, demonstrating G-alpha protein modification, stimulation of proliferation markers and expression of active β-catenin in a tissue- and cell-specific manner. This highlights the usefulness of PMT as an important tool for dissecting the specific roles of different G-alpha proteins in vivo.

## 1. Introduction

The bacterium *Pasteurella multocida* is often found as a commensal in the upper respiratory tract of many animals and birds [1]. Almost 70–90% of cats, and 25–50% of dogs carry the bacterium in the oropharyngeal region [2]. In humans, various studies have reported the association of *P. multocida* with rare but severe disease, often linked to infection from companion animals [3,4,5,6,7,8]. The bacterium is classified into the following serotypes based on the capsular antigen type A, Β, D, E, and F. Many type D, and some type A, strains of *P. multocida* contain the *toxA* gene encoding *Pasteurella multocida* toxin (PMT) [9]. Based on the presence or absence of *toxA*, the bacterium is classified to be toxigenic or non-toxigenic [9]. Toxigenic strains cause atrophic rhinitis in pigs, a condition associated with bone loss as well as systemic proliferative effects [10]. All these effects were shown to be caused by PMT, as they could be reproduced by using purified recombinant PMT [11]. In particular, we and others have suggested that the effects of PMT on bone remodelling in vivo are likely due to cell-autonomous effects on both bone-forming osteoblasts and bone-resorbing osteoclasts [12,13,14] The toxin has been identified as a potent mitogen that activates three families of heterotrimeric G-proteins—G_q/11_, G_12/13_, and G_i/o_—and thereby various signalling pathways downstream of these proteins [15,16,17,18,19,20,21,22,23,24,25,26,27], including β-catenin, where we have recently shown that PMT can regulate signalling in a G-alpha subunit-dependent manner [15,16,17,18,19,20,21,22,23,24,25,26,27]. These signalling interactions lead to increased mitogenesis [28,29,30] and other events at the cellular level, such as altered calcium signalling [31,32], activation of Rho GTPase pathways [12,33,34,35] and suppression of cyclic AMP signalling [28]. 

As PMT is one of the most potent mitogens known for eukaryotic cells—in the process, activating a myriad of procarcinogenic signalling pathways—we have previously suggested a potential role for PMT as a potential carcinogen [36,37], although how PMT acts in vivo is not well understood. Systemic effects of *Pasteurella multocida* were previously evaluated by us and others [38]. Either intraperitoneal injection with PMT or nasal infection with toxigenic *Pasteurella multocida* caused proliferation in the epithelium of bladder and ureter tissues [11,12,39,40]. Subcutaneous injection with PMT in rats was shown to induce weight loss and liver necrosis [41]. More recently, intraperitoneal injections with PMT in mice showed that the presence of PMT-modified G proteins in heart tissue and PMT also stimulated RhoA- and Rac1-mediated signalling in cultured cardiac cells [42]. In this study, we have further investigated the systemic effect of PMT by injecting the toxin into mice, and examining for the first time the spectrum of organs targeted by PMT using the QE antibody that specifically recognizes the PMT-modified G-proteins [26,43]. We also investigated the cellular effects of PMT in various tissues, namely stimulation of cell proliferation and active β-catenin, thus providing a more comprehensive map of the in vivo targets of PMT.

## 2. Results

### 2.1. Effects of PMT Treatment In Vivo 

To identify the targets of PMT in vivo, we treated mice for one week or one month with two 0.1 μg/kg intraperitoneal injections of PMT per week. After a one-week short-term treatment, we did not observe any significant differences in the weights of PMT-treated animals compared to either vehicle treatment or to an inactive mutant PMT (ΔPMT) that has no biological activity in vitro or in vivo [12,44] (Figure 1A(i),B(i)). However, there were indications after one week that animals treated with PMT exhibited a reduced rate of weight gain, and a longer one-month treatment with PMT with eight repeated intraperitoneal injections bi-weekly showed a significant reduction in the percent weight gained compared to the mice injected with either an inactive mutant PMT or vehicle control (Figure 1A(ii), B(ii)). The animals were otherwise healthy.

### 2.2. PMT Modifies G-Proteins In Vivo

To investigate whether PMT treatment caused the predicted G-alpha subunit modification in an in vivo context, we examined the presence of PMT-modified G-alpha proteins in individual organs obtained from PMT-treated mice. Western blot analysis using a QE antibody that specifically recognizes the glutamine (Q) to glutamic acid (E) modification induced by PMT [26] demonstrated that the expected 39 kDa PMT-modified G-proteins were detected in most organs analyzed after one month of PMT treatment, with organs such as the spleen, lungs, thymus, gonads, heart, bone and liver showing PMT modification of G-proteins as early as one week after injection (Figure 2). PMT-modified G-proteins were not observed following treatment with either vehicle control or inactive mutant PMT (Figure 2). We also did not observe modified G-proteins following PMT treatment in salivary glands, small intestine, brain and muscle tissue, at least within the one-month treatment period of the experiment (data not shown). These results characterize for the first time the spectrum of tissues that PMT can target directly in vivo.

### 2.3. PMT Treatment Induces Cell Proliferation Markers In Vivo

We next investigated whether PMT treatment caused any pathological and cellular changes within the one-month treatment period. In general, upon post-mortem dissection, the appearance at the macroscopic level of organs in PMT-treated mice did not show any overt phenotype, with the exception of a fraction of mice (2/4) that exhibited lesions on the ovaries resembling hemorrhagic cysts (Figure 3). This was the only gross pathology of note, at least within the one-month treatment period of the experiment.

We focused primarily on examining the mitotic indices that might be affected by PMT treatment, since we have previously shown that PMT can upregulate cell proliferation in vitro [18] and in vivo [39]. To this end, we investigated whether specific tissues that exhibited PMT modification of G-alpha subunits showed any evidence of PMT-induced mitotic activity through immunohistochemical analysis of the expression of the cell cycle markers Ki67 and phospho-Histone H3 (pHH3). In general, increased expression of both the pan cell cycle marker Ki67, and the mitotic phase marker pHH3, was observed in tissue sections of PMT-treated gonads, spleen and thymus, compared to control tissues (Figure 4). In uterine/endometrial tissue, both luminal and glandular epithelial cells showed mitotic activity, and PMT treatment showed an increase in Ki67 expression only in glandular epithelial cells, which was paralleled by increased expression of pHH3 (Figure 4A–D). In spleen tissue, PMT appeared to cause a marked increase in Ki67 staining in the red pulp with no apparent changes in the white pulp, and again, there was a concomitant increase in pHH3 staining in the red pulp (Figure 4E–H). In the thymus, the expression of Ki67-positive was greater within the cortex and pHH3 staining confirmed the marked increase in mitotic cells, especially within the outer peripheral regions of the thymic cortex (Figure 4I–L). The other organs that showed PMT-modified G-proteins did not show any obvious differences in mitotic figures compared to the controls, within the one-month treatment period (data not shown). These results demonstrate that PMT is biologically active when administered systemically and stimulates organ-specific cell proliferation in vivo.

### 2.4. PMT Treatment Stimulates Active β-Catenin In Vivo

We have previously demonstrated context-dependent regulation of active β-catenin by G-alpha subunits in vitro [27]. To evaluate the potential ability of PMT to regulate β-catenin in vivo, we performed immunohistochemistry for the expression of active β-catenin on the PMT-targeted organs with confirmed QE modification of target G-alpha proteins. In endometrial tissue, active membrane-bound β-catenin staining was observed in both luminal and glandular epithelial cells in both control and PMT-treated animals (Figure 5A–F). Upon closer examination, there appeared to be a higher proportion of uterine glands with altered morphology in PMT-treated tissue, containing smaller glandular epithelial cells and a correspondingly larger lumen size compared to control glands (Figure 5C,D). In addition, luminal epithelial cells in PMT-treated tissue showed active membrane-associated β-catenin localization in both apical and basal regions in contrast to control tissue which showed only apical expression (Figure 5E,F), although this expression pattern was not continuous throughout the epithelial lining. High expression of active β-catenin was also observed in sections of spleen and thymus. A marked increase in active β-catenin was observed in the red pulp of PMT-treated spleens which represents the majority of the splenic parenchyma (Figure 5G–J). Expression of active β-catenin was increased in PMT-treated thymus, specifically in the medulla as well as in cortical regions surrounding the connective tissue septae that demarcate the thymic lobules (Figure 5K–N). Thus, we have observed increased levels of active β-catenin in vivo in some organs that respond to PMT by G-alpha subunit modification.

## 3. Discussion

PMT treatment elicits a wide range of cellular effects in vitro, with the potential to have extensive sequelae in vivo. Although previous in vitro studies have provided detailed evidence about PMT-induced cell signalling, animal studies conducted either by injecting recombinant PMT or infecting animals with toxigenic strains of *Pasteurella multocida* have provided only limited analysis of PMT-induced organ damage, mainly based on gross pathological observation. Although bone tissue is a well-established target organ of PMT in vivo through its causal link to atrophic rhinitis [45,46,47], other studies have shown PMT-mediated proliferation of bladder epithelium in pigs, liver and kidney damage in pigs and rats, and hypertrophy of heart tissue in mice [3,39,41,42,48,49]. These studies suggest that there exist potential in vivo targets of PMT in addition to bone tissue. However, there remains a lack of comprehensive assessment of the spectrum of organs that could be sensitive to PMT action in vivo. At the molecular and biochemical levels, PMT function depends on the specific deamidation of glutamine (Q) to glutamic acid (E) in G-alpha proteins [43], and the development of an anti-QE antibody that is specific for deamidated G-alpha subunits provided a unique opportunity for identifying tissues that were *bona fide* direct targets for PMT in vivo [26]. Here, we have begun to identify these using the specific anti-QE antibody and have demonstrated that PMT has a wide range of in vivo targets leading to early signs of cellular changes.

As we were primarily interested in this study in identifying in vivo molecular targets of PMT rather than causing specific pathologies, we have chosen a relatively low dose of PMT, 5-fold lower than that used in previous studies that have shown to cause rapid, severe weight loss and organ toxicity/pathology [41]. We have performed our studies after a one-month treatment, after which mice were healthy, although the observed reduction in weight gain in PMT-treated animals suggests that exposure for longer periods might induce disease. Indeed, we have previously hypothesized that chronic exposure to PMT in vivo could drive pathological changes [37], and this is supported by our preliminary observations in vitro that long-term chronic treatment of cells with PMT over months changes their endogenous G-alpha subunit content (Banu, Lax, Grigoriadis, in preparation).

We have shown that PMT modifies G-proteins in multiple organs in vivo by confirming the specific Q > E modification biochemically. These tissue-specific PMT effects we observed suggest the involvement of a potential receptor required for its entry in different cell types. Cellular uptake of PMT occurs through its binding to sphingomyelin with a suggestion of associated protein receptors [50]. However, the precise receptors involved in PMT translocation still remain unclear. Hence, further exploring PMT tissue targets and non-targets could potentially reveal the receptors involved in PMT uptake. Nevertheless, in those tissues where we demonstrated clear deamidating effects of PMT, we observed increases in mitogenic activity as marked by positive Ki67 and pHH3 staining that are characteristic of PMT action [18]. Expression of these mitotic markers appeared to be specific within each tissue tested: glandular epithelial cells in the endometrium, the parenchyma in the red pulp of the spleen and the cells in the thymic cortex all appeared to be affected preferentially by PMT. Moreover, increased expression and localization of active β-catenin correlated well with the increases in Ki67 and pHH3 in the same subpopulations, particularly in endometrial and splenic tissues. Whether the observed effects of PMT on the proliferation markers and on active β-catenin expression are mediated by activation of specific G-alpha subunits in these tissues is not yet known. Moreover, whether the gonadal effect is also seen in the male reproductive system is not yet known but would be interesting as G_q_ expression is high in prostate and testis [51].

There were no obvious extreme pathological effects at the macroscopic level observed with PMT treatment, which may be due to the limited incubation period in this experiment and/or the intentionally low dose of PMT used. However, we did observe that a proportion of PMT-treated mice developed lesions resembling hemorrhagic cysts in the ovaries although the causes of this pathology are not clear. The increases in active β-catenin levels, however, are very interesting and relevant to our recent demonstration of a context-dependent role of PMT-modified G-alpha subunits in regulating active β-catenin signalling in the presence or absence of Wnt ligand [27]. β-catenin has been reported to be important for endometrial proliferation and differentiation, can regulate endometrial homeostasis and may show differential epithelial cell localization in proliferative and secretory phases of uterine function [52,53]. The altered membrane localization of β-catenin in PMT-treated luminal epithelial cells shows engagement with cadherins and altered apico-basal cell polarity, which is associated with tumour progression [54]. Finally, the increase in expression of active β-catenin in PMT-treated spleens is restricted to the parenchyma that is comprised of connective tissue cells which are known targets of PMT [18], and Wnt/β-catenin signaling is classically associated with fibrosis [55]. Thus, given that β-catenin has multiple cellular functions involving Wnt signalling activation, maintaining adhesion junctions, and acting as a transcription factor [56], it would be interesting in the future to further dissect the cell types in which β-catenin is activated and explore the functional role of the PMT-activated β-catenin in these tissues. Although our results require further functional validation of PMT action in an organ- and cell-specific manner, our study further intensifies the notion that PMT-mediated activation of G-alpha subunits is context dependent, and demonstrates that PMT is a valuable tool to dissect the tissue-specific roles of individual G-alpha subunits under normal and disease conditions in vivo using specific animal models.

## 4. Materials and Methods

### 4.1. Animal Studies

All the animal procedures were carried out according to the UK home office guidelines (Licence PPL 70/7866, approved 17/12/2013 by the Animal Welfare and Ethical Review Body (AWERB), King’s College London). Female CD1 mice were chosen for the experiments. Based on previous mouse studies, a dose of PMT (0.1 µg/kg of body weight) was chosen for injections. Mice (*n* = 4 per group) were treated with either PMT, an inactive PMT mutant (ΔPMT) containing a C-terminal cysteine to serine exchange (C1165S) [44], or vehicle (PBS) by intraperitoneal injections 2x per week for a total of either one week or one month. The weights of the mice were recorded at the beginning of the experiment and at culling. Various organs (brain, bone, skin, thymus, spleen, lungs, heart, liver, small intestine, colon, bladder, gonads, muscle, salivary glands, kidney) were harvested for biochemical analysis and histological examination. Inactive mutant ΔPMT did not exhibit any pathological, cellular or biochemical effects in vivo [41].

### 4.2. Histology

The extracted organs were fixed in 4% paraformaldehyde in PBS, processed and paraffin embedded. Sections (5 µm thickness) were cut and mounted on SuperFrost slides (Thermo Fisher Scientific, Hemel Hempstead, UK) for further evaluation. 

### 4.3. Tissue Homogenisation

Mouse tissue was snap frozen in liquid nitrogen and stored at −80 °C. Lysis buffer (25 mM Bicine pH 7.5, 150 mM NaCl, 2 mM Na_3_VO_4_, 1 mM NaF, 20 mM Na_4_P_2_O_7_, 0.02% NaN_3_, and proprietary detergent) at a final concentration of 5 mg/mL containing protease inhibitors (#P2714-1BTL, Sigma-Aldrich, Poole, UK) was added to the extracted organs. The tissue was homogenized using the lysing matrix D beads (#116913100, MP Biomedicals, Loughborough, UK) in the FASTPREP-24 system (MP Biomedicals), where the tissue along with lysis buffer was added in the tube containing specific beads and homogenized for 1 min, with 2 min for bone and muscle. The homogenate was transferred to a pre-chilled microfuge tube and incubated on ice on a rocking platform for 10 min. The tubes were spun at 14,000 RCF for 15 min and the supernatant was aliquoted and stored at −80 °C. 

### 4.4. Immunohistochemistry

The paraffin sections were dewaxed in xylene and rehydrated through a decreasing series of ethanol concentrations to water. The slides were then subjected to heat-induced antigen retrieval, using the sodium citrate buffer (pH 6) in a Decloaking chamber NXGEN (Menarini Diagnostics, Wokingham, UK), at 110 °C for 3 min. The slides were removed and washed in tap water. The sections were incubated in blocking solution containing 5% BSA in phosphate-buffered saline (PBS) for 30 min and washed with PBS. The sections were incubated with the primary antibody overnight in a humidified chamber at 4 °C. The following day, the sections were washed with PBS containing 0.01% Tween-20 (PBST) and incubated with 3% H_2_O_2_ solution for 15 min and rinsed thoroughly in tap water. The sections were incubated with secondary antibody for 2 h. The dilutions of both primary and the secondary antibodies were made in the blocking solution. After incubation, the slides were washed with PBST and then developed using the peroxidase substrate DAB kit (Vector labs, Peterborough, UK) as instructed by the manufacturer. Briefly, the developing solution was prepared by adding a drop of chromogen to 1 mL substrate, and the solution was applied to the section and incubated for 2 min. The sections were washed and counterstained with hematoxylin. They were then mounted using DPX and observed under the light microscope. 

### 4.5. Western Blot Analysis

Proteins were separated on 12% gradient bis-acrylamide gels using SDS-PAGE [57]. The proteins were electro-transferred onto nitrocellulose membrane using a transfer cell (Biorad, Hertfordshire, UK). Blots were incubated in 5% non-fat milk in Tris-buffered saline (TBS) for 30 min. Primary and secondary antibody dilutions were made in 5% non-fat milk in TBS. All the primary antibodies were used at a 1:1000 dilution with incubation overnight at 4 °C. The housekeeping protein β-actin at a 1:5000 dilution was incubated for 2 h at RT. Primary antibody was removed and the blots were washed three times for 10 min each with TBST (TBS with 0.1% Tween-20) and incubated with the respective HRP-bound secondary antibodies (1:5000 dilution) for 2 h at RT. Secondary antibody was removed and the blots were washed with TBST three times for 10 min each. The proteins were visualized using the chemiluminescence detector and the Chemidoc MP imaging system from Biorad.

## Figures and Tables

**Figure 1 ijms-21-02739-f001:**
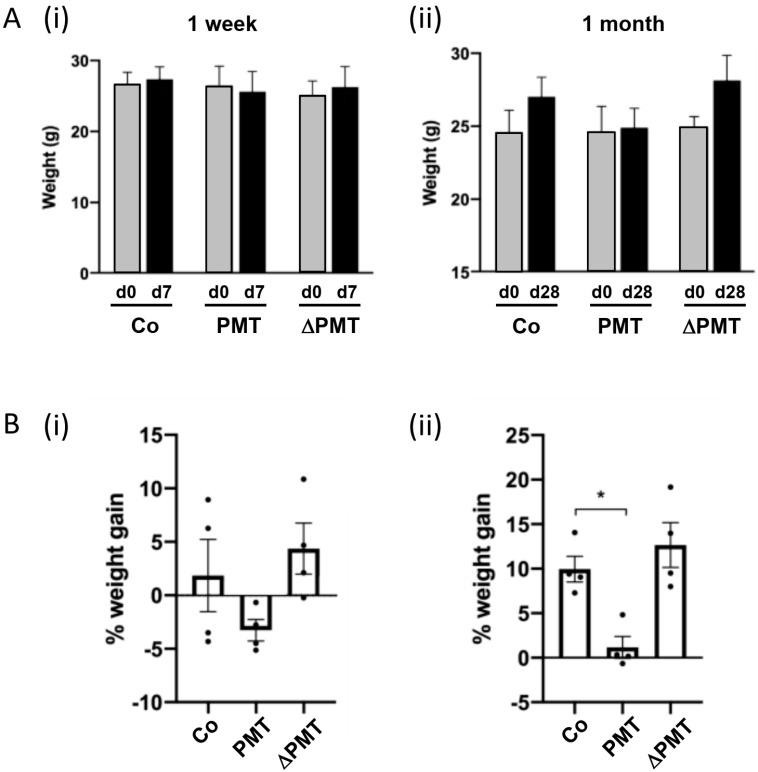
*Pasteurella multocida* toxin (PMT) treatment in vivo affects weight gain. Effects of short-term and long-term treatment of PMT in mice in vivo. Mice were treated with 3 ng recombinant PMT, inactive mutant PMT (ΔPMT; 3 ng), or PBS vehicle (Co) for either one week or one month as indicated. Body weights were measured and depicted either as (**A**) actual weights from days 0 to 7 (A(i), 1 week) and days 0 to 28 (A(ii), 1 month), or (**B**) percent weight gain within the 1 week (B(i)) and 1 month (B(ii)) periods. The data represent the mean ± SD (*n* = 4 per group) (* *p* < 0.05) (one-way Anova).

**Figure 2 ijms-21-02739-f002:**
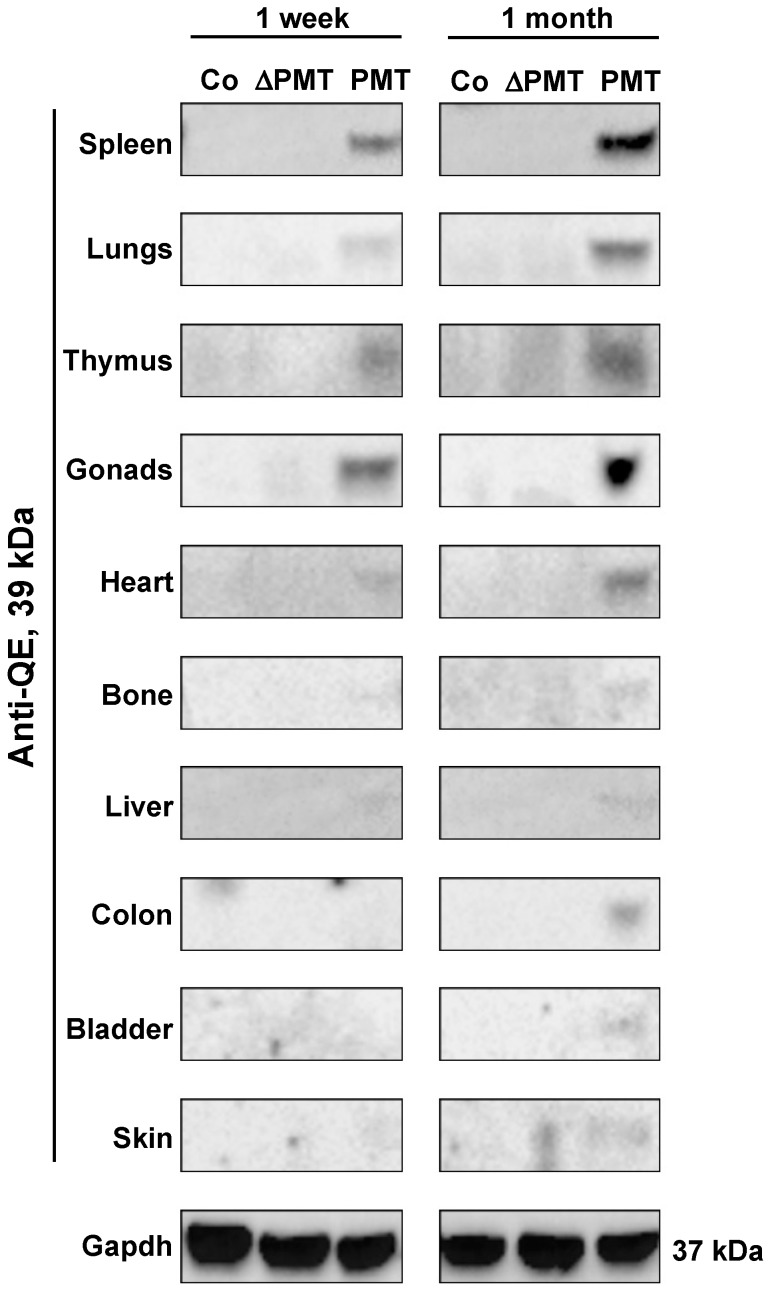
PMT modifies G-proteins in vivo. Western blot analysis of PMT-modified G-alpha protein subunits in whole-tissue extracts of mice treated with PMT (3 ng), mutant PMT (ΔPMT; 3 ng) or vehicle control (Co) for either one week or one month as indicated. Extracts were immunoblotted with a specific anti-QE antibody as described in the Methods. Gapdh was used as a loading control.

**Figure 3 ijms-21-02739-f003:**
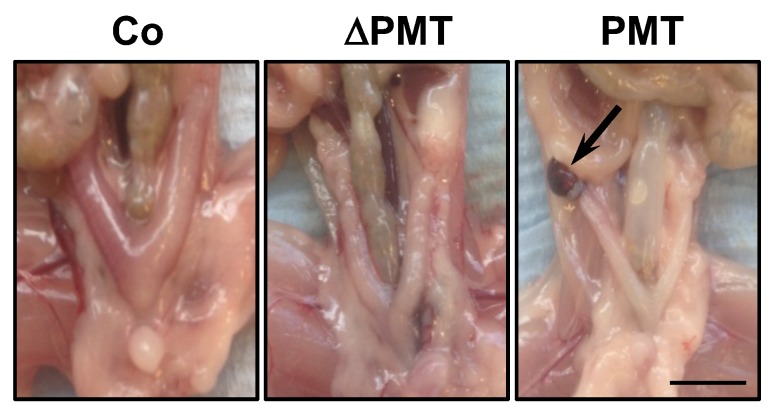
Gross pathology of PMT-treated mice. Representative dissections of mice treated with PMT (3 ng), mutant PMT (ΔPMT; 3 ng) or vehicle control (Co) for one month showing a lesion resembling a hemorrhagic cyst (arrow) in PMT-treated mice. Scale bar: 5 mm.

**Figure 4 ijms-21-02739-f004:**
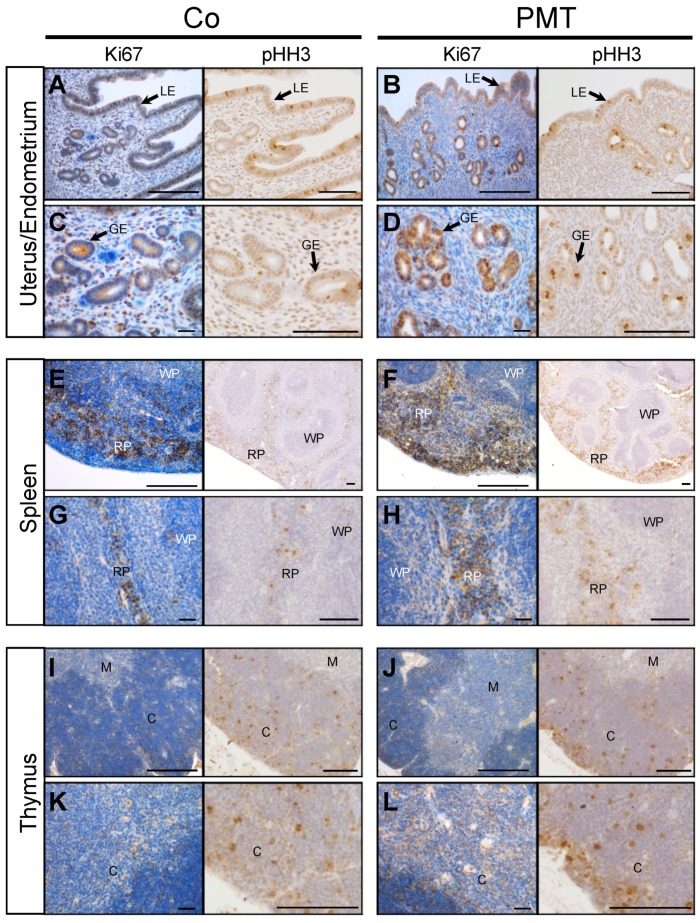
PMT stimulates proliferation markers in vivo. Immunohistochemical analysis of the cell cycle marker, Ki67, and mitosis marker, pHH3, in uterine (**A**–**D**), spleen (**E**–**H**) and thymus (**I**–**L**) tissues of animals treated with either PMT (3 ng) or vehicle control (Co) for one month. Micrographs represent 5 μm paraffin sections stained with specific antibodies against Ki67 or phospho-Histone H3 (pHH3) as described in the Methods. Uterus/endometrium: luminal epithelium (LE), glandular epithelium (GE); spleen: red pulp (RP), white pulp (WP); thymus: cortex (C), medulla (M). Scale bars: 100 μm.

**Figure 5 ijms-21-02739-f005:**
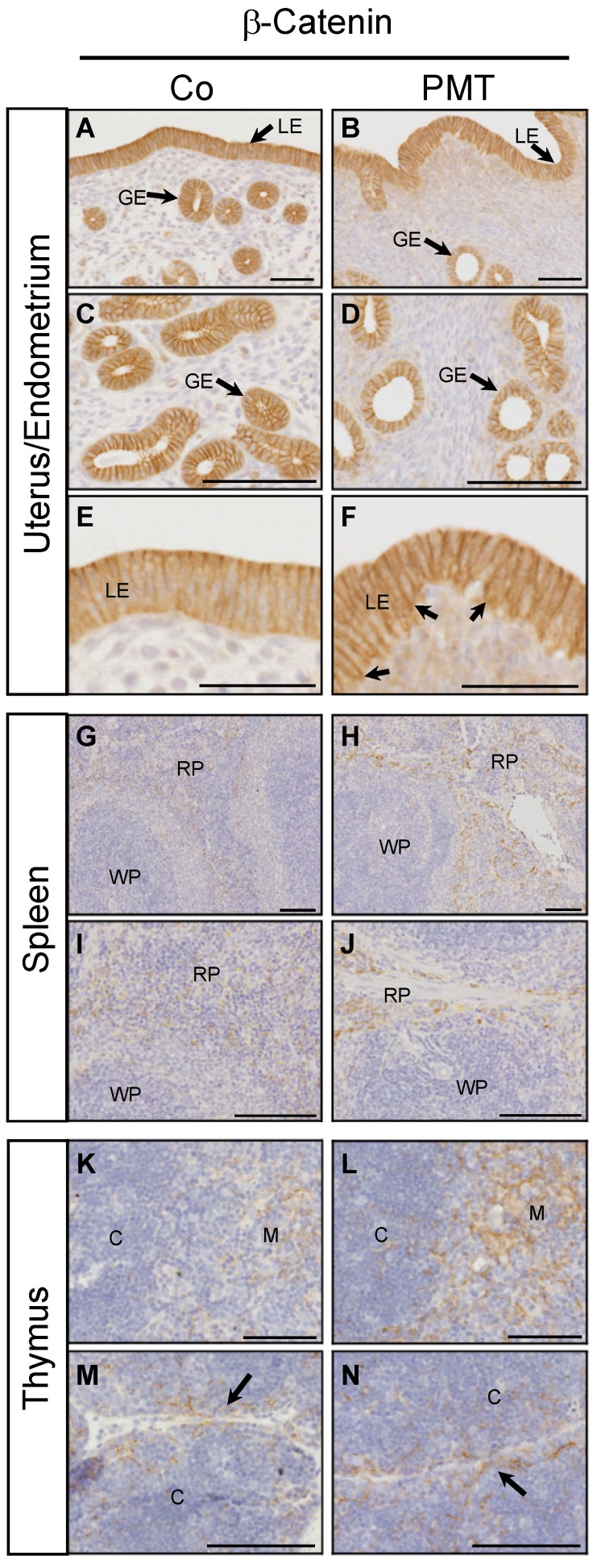
PMT stimulates active β-catenin in vivo. Immunohistochemical analysis of active β-catenin in uterine (**A**–**F**), spleen (**G**–**J**) and thymus (**K**–**N**) tissues of animals treated with either PMT (3 ng) or vehicle control (Co) for one month. Micrographs represent 5 μm paraffin sections stained with specific antibodies recognizing active β-catenin as described in the Methods. Uterus/endometrium: luminal epithelium (LE), glandular epithelium (GE); spleen: red pulp (RP), white pulp (WP); thymus: cortex (C), medulla (M). Arrows in F indicate increased basal epithelial membrane staining of β-catenin in LE cells. Arrows in (**M**,**N**) indicate the connective tissue septae that demarcate the thymic lobules. Scale bars: (A–D)(G–N): 100 μm; (E,F): 50 μm.

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
