# Peer review of "In Vivo Targets of *Pasteurella Multocida* Toxin"

_ijms, 2020, doi:10.3390/ijms21082739_

Round 1

Reviewer 1 Report

Authors investigated effects of the Pasteurella multocida toxin (PMT) in mice. They performed in vivo study on mice treated with PMT for a limited time (up to 1 month) and using relatively low PMT dose, to capture initial PMT effects.  For the study, they exploited the QE antibody recognizing alpha subunit of G proteins modified by PMT mediated deamidation of glutamin (Q→E).  In vivo experiments following author´s previous in vitro studies, show that PMT targets multiple organs and tissues.  Authors detected stimulation of proliferation markers and expression of active b-catenin in tissues targeted by PMT.   

This, well written manuscript represents a useful but just preliminary study to address many really interesting and essential questions on the molecular basis of the mechanisms of action of PMT, changes in G protein function, as well as the identification of the PMT receptor and its effect on PMT targetting. 

I have no objections to the presentation of the data or to the processing of the manuscript.

Author Response

1. I have no objections to the presentation of the data or to the processing of the manuscript.

Response 1:  We thank the Reviewer for acknowledging that despite the early stages and descriptive nature of the data presented, the manuscript provides useful novel information about the consequences of PMT targeting in vivo.

Reviewer 2 Report

The studies, as presented, are well executed and, a nicely complete in vivo package. The paper makes a relevant contribution to the field and is highly suitable for inclusion in International Journal of Molecular Sciences.

A few suggestions to improve the paper which I hope the authors will note are:

  • In figures 1, 2, 3 the delta symbol is missing (at least 7 times; delta-PMT); for figure 5 beta symbol is not present (beta-catenin)
  • The reference Orth et al 2009 PNAS should be included in the manuscript (demonstration of deamidase activity)

Given the activity on gonads, it may be interesting in future experiments to study effects of PMT on male mice.

Author Response

We thank the Reviewer for the positive response and for acknowledging that the paper makes a significant contribution to the field.

Specific responses:

1. In figures 1, 2, 3 the delta symbol is missing (at least 7 times; delta-PMT); for figure 5 beta symbol is not present (beta-catenin)

Response 1: We apologise for this typographical error.  All the Greek symbol fonts appeared to be correct in the submitted version, so something must have occurred in the file that was uploaded to the Reviewers.  We will ensure with the Journal and Copy Editor that the final version is correct.

2. The reference Orth et al 2009 PNAS should be included in the manuscript (demonstration of deamidase activity)

Response 2:  Many thanks for bringing this to our attention.   Absolutely, this reference must be included and we have done so in the revised manuscript.

3. Given the activity on gonads, it may be interesting in future experiments to study effects of PMT on male mice.

Response 3:  This is a very interesting point and we have added a sentence and references to this effect, as the gonads and prostate of the male reproductive system are also know to contain high levels of Gq.